# Oral Microbiota Signatures in the Pathogenesis of Euthyroid Hashimoto’s Thyroiditis

**DOI:** 10.3390/biomedicines11041012

**Published:** 2023-03-26

**Authors:** Mustafa Genco Erdem, Ozge Unlu, Fatma Ates, Denizhan Karis, Mehmet Demirci

**Affiliations:** 1Department of Internal Medicine, Faculty of Medicine, Beykent University, Istanbul 34398, Turkey; 2Department of Medical Microbiology, Faculty of Medicine, Istanbul Atlas University, İstanbul 34403, Turkey; 3Department of Biophysics, Faculty of Medicine, Beykent University, Istanbul 34398, Turkey; 4Department of Biophysics, Faculty of Medicine, Istinye University, Istanbul 34460, Turkey; 5Department of Medical Microbiology, Faculty of Medicine, Kirklareli University, Kirklareli 39100, Turkey

**Keywords:** autoimmunity, Hashimoto’s thyroiditis, oral microbiota

## Abstract

One of the most prevalent autoimmune illnesses in the world is Hashimoto’s thyroiditis, whose pathogenesis is still unknown. The gut–thyroid axis is frequently examined, and although oral health affects thyroid functions, there are limited data on how oral microbiota is linked to Hashimoto’s thyroiditis. The study aims to identify the oral microbiota from saliva samples taken from treated (with levothyroxine) and untreated female euthyroid Hashimoto’s thyroiditis patients as well as healthy controls who were age- and sex-matched to compare the oral microbiota across the groups and to contribute preliminary data to the literature. This study was designed as a single-center cross-sectional observational study. Sixty (60) female patients with euthyroid Hashimoto’s thyroiditis (HT) and eighteen (18) age- and gender-matched healthy controls were included in this study. Unstimulated saliva samples were collected. After DNA isolation, sequencing was performed by targeting the V3-V4 gene regions of the 16S rRNA on the MiSeq instrument. R scripts and SPSS were used for bioinformatic and statistical analysis. No significant differences were found in the diversity indices. However, *Patescibacteria* phylum showed a significantly higher abundance (3.59 vs. 1.12; *p* = 0.022) in the oral microbiota of HT patients compared to HC. In the oral microbiota, the euthyroid HT group had approximately 7, 9, and 10-fold higher levels of the *Gemella, Enterococcus*, and *Bacillus* genera levels than healthy controls, respectively. In conclusion, the results of our study demonstrated that Hashimoto’s thyroiditis causes changes in the oral microbiota, whereas the medicine used to treat the condition had no such effects. Therefore, revealing the core oral microbiota and long-term follow-up of the HT process by conducting extensive and multicenter studies might provide some important data for understanding the pathogenesis of the disease.

## 1. Introduction

Hashimoto’s thyroiditis is a common autoimmune disease. Environmental factors such as microbiota composition and genetic factors cause an imbalance in the self-tolerance mechanism, which must be regulated by the immune system’s regulatory T and B lymphocytes. Nutritional habits may also affect the natural course of Hashimoto’s thyroiditis [1]. The pathogenesis of Hashimoto’s thyroiditis is still unclear [2]. The thyroid gland is the organ thought to be most affected by autoimmune processes. Due to an imbalance in the self-tolerance systems, lymphocyte infiltration damages the thyroid gland [1,3]. Furthermore, it is well-known that women are 4–10 times more likely than men to develop Hashimoto’s thyroiditis [4]. HT was described by Hakaru Hashimoto in 1912, but it was not until the 1950s that it was recognized as an autoimmune disease. It is thought that HT is caused by a problem with the immune system that results from the interaction of genetics and the environment. In the morphology of HT, follicular atrophy and hyperemia accompanied by oncocytic metaplasia of follicular cells and gradual atrophy of thyroid tissue following the invasion of the thyroid gland by lymphocyte cells should be considered. This process leads to hypothyroidism in the thyroid gland in patients with HT; although some patients with HT may have normal thyroid gland function. The dysfunction of the thyroid gland can be differentiated depending on the extent of the damaged thyroid gland parenchyma. Bossowski and Otto-Buczkowska reported that serologically, antithyroid antibodies, especially thyroperoxidase (anti-TPO) and anti-thyroglobulin (anti-Tg), positively correlate with an increased inflammatory response in the thyroid and hypothyroidism [5]. At the cellular level, HT is an autoimmune process that leads to cell-mediated immunity with increased Th1 lymphocyte activity and subsequent death of the host thyroid gland cell. In addition to Th1, Th17 is reported to play a role in the immunopathogenesis of HT and cause autoimmunity of the thyroid gland. Th17 lymphocytes have been shown to be responsible for developing and exacerbating the chronic inflammatory response that occurs in most autoimmune diseases such as HT. It is also thought that there is a balance between regulatory T lymphocytes (Treg) and Th17 lymphocyte levels in patients with HT. These Treg lymphocytes also show dysfunction in the immunopathogenesis of many autoimmune diseases, such as HT [6]. The oral microbiota can be defined as the collective collection of microorganisms present in the oral cavity. After the gut microbiota, the oral microbiota contains the second-largest microbial community in humans [7]. The oral microbiota includes the phylums *Actinobacteria*, *Bacteroidetes*, *Chlamydia*, *Euryarchaeota*, *Fusobacteria*, *Firmicutes*, *Proteobacteria*, *Spirochaetes*, and *Tenericutes* and is known to contain approximately 1000 different bacterial species [8]. Studies on oral microbiota have shown that oral microbiota-derived bacteria can be found in the intestines of patients with different diseases. It is unclear whether bacteria in the oral microbiota can induce inflammation in the intestines and cause systemic diseases. However, studies have shown that bacteria in the oral microbiota can colonize the intestines, leading to activation of the immune system through the intestines and chronic inflammation in systemic diseases. Saliva samples from patients with inflammatory diseases such as Crohn’s disease transferred to germ-free mice can cause a marked increase in Th1 lymphocytes in the intestinal lamina propria of these mice [9]. Th17 cells are known to have a physiological and protective role in the development of the immune response of the oral mucosa. In both human and mouse models, it has been shown that defects in Th17 lymphocyte cells and IL-17 cytokine signaling can predispose to oral fungal infections. All patients with an immunodeficiency in which the differentiation or function of Th17 lymphocytes is affected are susceptible to oral infections. In oral cavity infections such as periodontitis, Th17 lymphocytes have been shown to play an important role in shaping the immune response leading to increased development of gingivitis and tooth loss [10]. Tolerogenic dendritic cells (DC) are found in the oral mucosa. These cells produce immunomodulatory cytokines such as IL-10, TGF-β1, and Prostaglandin E2 in the oral mucosa and are in close cross-talk with oral mucosal Tregs lymphocyte. The presence of oral mucosal Foxp3+ Tregs with protective functions during local infection has been reported. A reciprocal relationship between these regulatory cells and the oral microbiota has been observed during oral infection. During the early acute response in the oral mucosa, Treg cells are thought to enhance Th17 cell responses and neutrophil infiltration [11]. The background regulatory mechanism and how microorganisms induce the Th17 cell-mediated immune response in the oral microbiota are unclear. An oral–gut microbiota axis has been proposed, through which oral microbiota-derived bacteria have been shown to exacerbate systemic inflammatory diseases such as colitis or obesity via Th17 lymphocytes. In addition, it has been emphasized how the oral–gut microbiota axis and the immune response network can integrate into multi-organ interactions [12]. The gut microbiota is considered an important environmental factor in disease development. Patients often report changes in their quality of life and thyroid functions as a result of dietary changes [3,13]. It is thought that the gut microbiota and its metabolites, such as short-chain fatty acids, have an important effect on the normal functioning of the thyroid gland [14]. Patients with hypothyroidism associated with Hashimoto’s thyroiditis usually require lifelong hormone replacement therapy with levothyroxine [3]. It is also known that the gut microbiota can affect the levels of thyroid hormones by regulating iodine uptake, enterohepatic recycling of iodine, and levothyroxine bioavailability. Due to these factors, researchers studying the gut–thyroid axis are typically interested in autoimmune thyroid illnesses, such as Hashimoto’s thyroiditis [14,15]. The beginning of the digestive system, the oral microbiota, is essential for controlling the immune system. It is well-known that various disorders cause alterations in the oral microbiota [16]. In a population-based cross-sectional study, it was found that oral health affects thyroid functions. However, it has been reported that more studies are needed to determine the causal relationship [17]. Oral microbiota may interact with thyroid illnesses due to mouth-to-gut and gut-to-mouth microbial bidirectional interactions, which can alter the etiology of numerous diseases [18]. Very few studies examine the oral microbiota in Hashimoto’s thyroiditis patients [19]. The purpose of this study was to identify the oral microbiota from saliva samples collected from female patients with euthyroid Hashimoto’s thyroiditis who were not receiving treatment and from those who were receiving levothyroxine (L-T4) treatment, as well as from healthy controls, and to compare the oral microbiota between the groups as well as to provide preliminary data to the literature.

## 2. Materials and Methods

### 2.1. Study Design and Participants

This study was designed as a single-center cross-sectional observational study. All participants gave their informed consent, and the Istinye University Ethical Committee (2/2021.K-43) approved the study. The exclusion criteria of our study were I. Surgical operation on the salivary glands; II. Pregnancy; III. Fixed orthodontic treatment; IV. Antibiotic, anti-inflammatory, and anticoagulant therapy or probiotic use for 4 weeks; V. Non-adults (<18 years of age); VI. Active viral, bacterial, or fungal infection; VIII. Presence of any active oral diseases.

Sixty (60) female patients with euthyroid Hashimoto’s thyroiditis (HT) and eighteen (18) age- and gender-matched healthy controls were included in this study. HT diagnosis was based on the serum thyroid peroxidase antibody (TPOAb or anti-TPO) levels, and the thyroid floor parenchyma echogenicity was heterogeneous; the structure was coarsened according to thyroid ultrasonography results. All participants’ demographic information and medical records were checked from the electronic health records systems in the hospital before the sample collection. The HT patient group was divided into two subgroups: These were twenty-eight (28) female euthyroid HT treated with levothyroxine (L-T4) (LT + HT) subgroup and thirty-two (32) female euthyroid HT without L-T4 therapy (LT-HT) subgroup [20].

### 2.2. Sample Collection and Storage

Unstimulated saliva samples were randomly collected from all participants admitted to the internal medicine clinics. The study participants were informed that they should brush their teeth at least 1 h before the sampling and should not consume cigarettes, drinks, or food until the sampling. Participants were asked to rinse their mouths with 0.9% saline for 1 min. Then, 5 min after mouth washing, at least 5 mL of unstimulated saliva samples were taken into 50 mL sterile tubes. The sample was quickly transported to the laboratory in appropriate conditions. DNA/RNA Shield reagent (Zymo Research Corp, Irvine, CA, USA) was added and stored at −80 °C until the nucleic acid extraction [21].

### 2.3. DNA Extraction from Unstimulated Saliva Samples

DNA was extracted from unstimulated saliva samples using Quick-DNA Fecal/Soil Microbe Kits (Zymo Research, CA, USA) according to the manufacturer’s instructions. The concentration and purity of DNA samples were evaluated by Qubit (Thermo Fisher Scientific, Waltham, MA, USA) before sequencing. 

### 2.4. PCR Amplification and Sequencing

Next-generation sequencing was performed by targeting the V3-V4 gene regions of the 16S rRNA. 16S Universal Eubacterial primers (16S Forward: TCG TCG GCA GCG TCA GAT GTG TAT AAG AGA CAG CCT ACG GGN GGC WGC AG; 16S Forward Reverse: GTC TCG TGG GCT CGG AGA TGT GTA TAA GAG ACA GGA CTA CHV GGG TAT CTA ATC C) were used for amplification [22]. During the library preparation, a two-step PCR was performed. In these procedures, 25 cycles of PCR were performed separately for each sample using KAPA HiFi HotStart ReadyMix (Roche Diagnostics GmBH, Mannheim, Germany). In the first PCR step, the PCR condition was 3 min at 95 °C, then 30 s at 95 °C, 30 s at 55 °C, 30 s at 72 °C for 25 cycles, and, finally, a single cycle at 72 °C for 5 min. In the second PCR application, Nextera XT Index Primer 1 and Nextera XT Index Primer 2 sets (Illumina, San Diego, CA, USA) were added to the Illumina index and adapter sequences. In this PCR step, the PCR condition was 3 min at 95 °C; 30 s at 95 °C, 30 s at 55 °C, 30 s at 72 °C for 8 cycles; and then a single cycle at 72 °C for 5 min. After both PCR steps, purification of the amplicon products was performed with the Agencourt AMPure XP kit (Beckman Coulter, Fullerton, CA, USA) [23]. After the PCR steps, the PCR products of all samples were checked for band presence and relative band intensities on a 2% agarose gel. The prepared library was measured with a Qubit fluorometer and sequenced after normalization. Sequencing was performed on the MiSeq instrument (Illumina, San Diego, CA, USA) following the paired-end 2 × 300 bp protocol according to the manufacturer’s recommendations [24]. 

### 2.5. Bioinformatics and Statistical Analysis

The FASTQC tool was used to check the quality of the read data after sequencing. According to the quality control results, data quantities, read qualities, GC distributions, kmer distributions, and possible adapter contaminations were analyzed for each sample. During the sequencing process, low-quality base reads and possible adapter-index contaminations in the raw read data were trimmed from the reads to avoid biases in subsequent analysis steps. The Trimmomatic (v0.39) tool was used for quality trimming. In this step, low-quality base reads, possible adapter contaminants, and chimeric sequences in the read ends were trimmed using the Trimmomatic tool based on the Genomes OnLine Database (GOLD). For taxonomic profiling, reads were aligned to target organisms based on the SILVA (2022) database using the Kraken2 tool. After alignment, OTU groups in each sample were identified. The R::vegan package was used to calculate diversity indices. R scripts were used for data reporting, statistical analysis, and data visualization [25]. The IBM SPSS version 20 software used demographic results from medical records. Data are given as number (n), mean, and standard deviation (SD). Comparisons between groups were analyzed using Kruskal Wallis tests. *p* < 0.05 value was considered significant in all analyses.

## 3. Results

There was no statistically significant difference between the euthyroid Hashimoto’s thyroiditis female patients included in our study and healthy controls regarding age, BMI, and TSH levels (*p* > 0.05). The demographic characteristics of the participants are given in Table 1.

An average of 50.305 reads per sample (SD = 6756; range: 43.549–57.061) were obtained. The length of clean reads ranged from 422 to 439 bp. At the 97% similarity level, all reads were clustered into 70 phyla, 162 classes, 372 orders, 583 families, and 1496 genera. No significant differences were found in the alpha-diversity indices such as Shannon or Simpson (Table 2). In the beta-diversity, no significant cluster separations were also found using with principal coordinate analysis (PCoA) between groups.

Twelve (12) phyla and one hundred and sixty-three (163) genera showed significant differences (*p* < 0.05) between the HT and HC groups. The Venn diagram shows the phylum showing a significant difference between the healthy control, LT – HT and LT + HT groups (Figure 1).

In relative abundance at the phylum levels, *Firmicutes* (39.52 vs. 33.93), *Bacteroidota* (17.38 vs. 12.57), *Actinobacteriota* (13.4 vs. 10.19), *Proteobacteria* (4.05 vs. 17.71), *Patescibacteria* (3.59 vs. 1.12), *Fusobacteriota* (2.78 vs. 2.6), *Spirochaetota* (0.25 vs. 0.13), and *Campylobacterota* (0.14 vs. 0.42) were the dominant phyla in the taxonomic composition of saliva samples. Among these, only *Patescibacteria* showed a statistical difference between groups. A significantly higher abundance (3.59 vs. 1.12; *p* = 0.022) of this phylum was found in HT than in healthy controls (Figure 2a). There were no statistically significant differences between LT-HT and LT + HT groups (Figure 2b).

The most dominant genera in the saliva samples of all participants are shown in Figure 3.

In relative abundance at the genus levels, *Streptococcus* (3.42 vs. 2.16), *Prevotella*_7 (1.3 vs. 1.46), *Porphyromonas* (1.24 vs. 0.24), Actinomyces (0.79 vs. 0.84), *Gemella* (0.71 vs. 0.11), *Rothia* (0.68 vs. 0.24), *Peptostreptococcus* (0.44 vs. 0.1), *Veillonella* (0.44 vs. 1.4), *Prevotella* (0.43 vs. 0.46), and *Leptotrichia* (0.36 vs. 0.18) were the top 10 dominant genera in the taxonomic composition of saliva samples. Among these, only *Gemella* showed a statistical difference between groups. A significantly higher abundance (0.71 vs. 0.11; *p* < 0.001) of this genus was found in HT than in HC (Figure 4a). *Enterococcus* (0.099 vs. 0.014; *p* < 0.0001), *Staphylococcus* (0.08 vs. 0.01; *p* < 0.0001), and *Bacillus* (0.07 vs. 0.007; *p* < 0.0001) genera also showed a statistical difference between HT and HC. A significantly higher abundance of these genera were found in HT than in HC. There were also no statistically significant differences between LT-HT and LT + HT groups (Figure 4b).

*Gemella* (0.75 vs. 0.1; *p*:0.004), *Enterococcus* (0.098 vs. 0.014; *p* < 0.0001), *Staphylococcus* (0.097 vs. 0.013; *p* < 0.0001), and *Bacillus* (0.083 vs. 0.007; *p*: 0.004) were dominant genera and also showed a statistical difference between LT-HT and HC. *Gemella* (0.67 vs. 0.1; *p*: 0.048), *Enterococcus* (0.1 vs. 0.014; *p*: 0.02), and *Bacillus* (0.066 vs. 0.007; *p*: 0.004) genera also showed a statistical difference between LT + HT and HC, but *Staphylococcus* (0.065 vs. 0.013; *p*:0.07) genus did not show a statistical difference between LT + HT and HC. 

Approximately 7, 9, and 10-fold increases in *Gemella*, *Enterococcus*, and *Bacillus* genera, respectively, were statistically significant in all euthyroid HT patient groups compared to healthy controls, regardless of drug use.

## 4. Discussion

Hashimoto’s thyroiditis (HT) is one of the most common autoimmune diseases worldwide. It is thought that the entire thyroid balance may be sensitive to changes in microbiota composition. Therefore, evidence that the occurrence and progression of autoimmune thyroid disorders are particularly associated with altered gut microbiota composition is of interest [26]. Oral microbiota have been associated with different systemic diseases. It is known that bacteria in the oral microbiota can also pass into the intestine via saliva, change the composition of the gut microbiota, and affect the immune system [27,28]. Although oral microbial dysbiosis has been detected in all systemic autoimmune diseases, studies on the oral microbiota and autoimmune diseases are limited [29]. Thus, this study focused on identifying the oral microbiota from saliva samples taken from euthyroid female patients with Hashimoto’s thyroiditis who were receiving levothyroxine (L-T4) treatment and from those who were not, as well as from healthy controls, comparing the oral microbiota between the groups, and contributing preliminary data to the literature.

It has been determined that there are very limited studies on the connection between Hashimoto’s thyroiditis and the oral microbiome [19]. Similar to our research, Wang et al. found no significant differences in oral microbiome diversity between pregnant women with hypothyroidism and healthy controls. However, they found that the phyla *Firmicutes, Leptotrichiace*, and *Actinobacteria* decreased compared to healthy controls, whereas *Gammaproteobacteria* and *Prevotella* increased [19]. In our investigation, we discovered that although the alteration in the *Proteobacteria* phylum was identical, the *Firmicutes, Leptotrichiace*, and *Actinobacteria* phyla increased in individuals with euthyroid HT. We hypothesized that this was caused by the patient group in Wang et al., consisting of pregnant women with hypothyroidism.

Although the oral microbiota was the focus of our investigation, studies of the gut microbiota appear to be more common in HT patients [30,31,32]. We attempted to compare our results with similar studies because it is believed that the oral microbiota may also enter the gut through saliva and alter the gut microbiota composition [27,28]. Zhao et al. reported that they did not detect a significant difference in diversity in the gut microbiota in HT patients compared to healthy controls [30]. They reported that *Faecalibacterium*, Bacteroides, Prevotella_9, and *Lachnoclostridium* genera decreased [30]. However, the data of our study are not similar except that *Faecalibacterium. Bacteroides*, Prevotella_9, and Lachnoclostridium genera were higher in the oral microbiota of HT patients than in healthy controls. When Ishaq et al. [31] again compared the gut microbiota of HT patients with healthy controls, they reported a decrease in *Prevotella*_9 and *Dialister* genera and an increase in *Escherichia-Shigella* and Parasutterella genera [31]. Our results regarding the *Dialister* genus support the data of this study.

Contrary to our results, Ishaq et al. reported a decrease in *Firmicutes*, *Bacteroidetes*, *Actinobacteria*, *Verrucomicrobia*, and *Fusobacteria*; and an increase in *Proteobacteria* when their data were analyzed at the phylum level [31]. When Liu et al. compared the gut microbiota of euthyroid HT and hypothyroid HT patients using levothyroxine with healthy controls, they reported that euthyroid HT patients showed a more similar bacterial pattern with healthy controls [32]. In this study, they found an increase in *Bacteroidetes*, *Actinobacteria*, *Fusobacteria*, and *Proteobacteria* in the gut microbiota and a decrease in the relative abundance of *Firmicutes* and *Verrucomicrobia* at the phylum level. Similar to our study, they reported an increase in *Alistipes* and a decrease in *Faecalibacterium* at the genus level [32]. There appear to be differences even in the gut microbiota study data [30,31,32] performed on patients with HT. For this reason, the difference between the gut and oral microbiota might cause the differences observed in the data of our study. It is known that *Faecalibacterium prausnitzii* induces anti-inflammatory cytokines and suppresses the increase in pro-inflammatory cytokines [33]. For this reason, the similarity of the decrease in the abundance of *Faecalibacterium* in the patients’ oral or gut microbiota was striking, suggesting this decrease may also play a role in the mechanism of HT disease. Zhoa et al. reported that *Proteobacteria* and *Actinobacteria* were higher in the gut microbiota of HT patients compared to controls, and *Erysipelotrichia*, *Cyanobacteria*, and *Ruminococcus*_2 were higher in both HT and Graves patients compared to the control group [34]. Increases in the relative abundance of Actinobacteria were similar to our results. Table 3 summarizes different microbiota reports on HT and is presented for comparison with our study results.

In our study, a significant increase was found in the *Patescibacteria* phylum in the oral microbiota of the patients. Similar to the results of our study, Wang et al. detected an increase in this phylum in the oral microbiota of patients with primary Sjögren’s syndrome [35]. In addtion, Russell et al. reported that the increase in the relative abundance of the *Patescibacteria* phylum in the human gut microbiota is a genetic risk for autoimmunity [36]. In our study, a significant increase was found in the oral microbiota of the patients in the *Gemella* genus compared to the controls. Bruserud et al. found an increase in the oral microbiota of patients with autoimmune polyendocrine syndrome type 1 in the genus *Gemella*, supporting our data [37]. Another intriguing statistic was the *Enterococcus* genus, which revealed a significant increase in the oral microbiota of the patients in our study compared to the healthy controls. Bagavant et al. reported that *Enterococcus* galinarum antibodies might contribute to a more severe autoimmune process in the presence of a specific genetic background in systemic lupus patients, and this bacterium should be considered a pathobiont [38]. 

The limitations of this study are the single center design, the inability of our oral microbiota data to be validated by data from metabolomics or transcriptomics due to financial constraints, short-term patient follow-up, and the lack of data on the history of the disease. On the other hand, our study is significant since there is little information in the literature about the connection between oral microbiota and autoimmunity. We explore the oral microbiota, which influences other microorganisms, directly and indirectly, to contribute to the host autoimmune process.

## 5. Conclusions

The results of our study indicated that the oral microbiota was altered in Hashimoto’s thyroiditis, whereas the levothyroxine usage did not affect the oral microbiota. Changes in the *Gemella*, *Enterococcus*, *Faecalibacterium* genera, and *Patescibacteria* phylum have all been linked to the development of autoimmune diseases. On the other hand, more comprehensive and controlled clinical studies are required to determine whether the oral microbiota changes due to the disease, or whether the disease develops due to the change in the oral microbiota. These studies might provide essential data for detecting HT pathogenesis, extracting the core oral microbiota, and conducting long-term follow-up of the disease process.

## Figures and Tables

**Figure 1 biomedicines-11-01012-f001:**
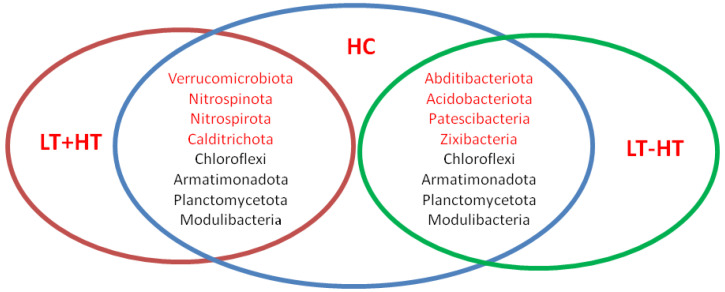
Phyla showing a significant difference between groups. (HT: Hashimoto’s thyroiditis. LT + HT: Euthyroid HT treated with levothyroxine (L-T4). LT-HT: euthyroid HT without L-T4 therapy. HC: Healthy control).

**Figure 2 biomedicines-11-01012-f002:**
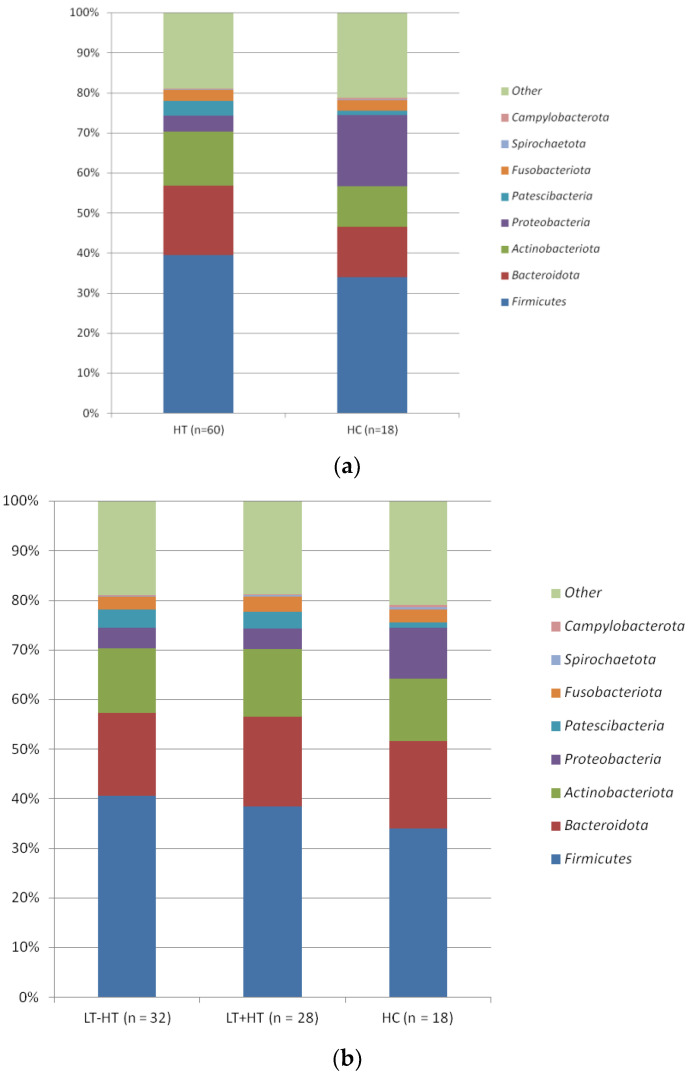
(**a**) The distributions of the predominant bacteria at the phylum level between HT and HC groups. (HT: Hashimoto’s thyroiditis. HC: Healthy control). (**b**) The distributions of the predominant bacteria at the phylum level between LT-HT, LT + HT, and HC groups (HT: Hashimoto’s thyroiditis. LT + HT: Euthyroid HT treated with levothyroxine (L-T4). LT-HT: euthyroid HT without L-T4 therapy. HC: Healthy control).

**Figure 3 biomedicines-11-01012-f003:**
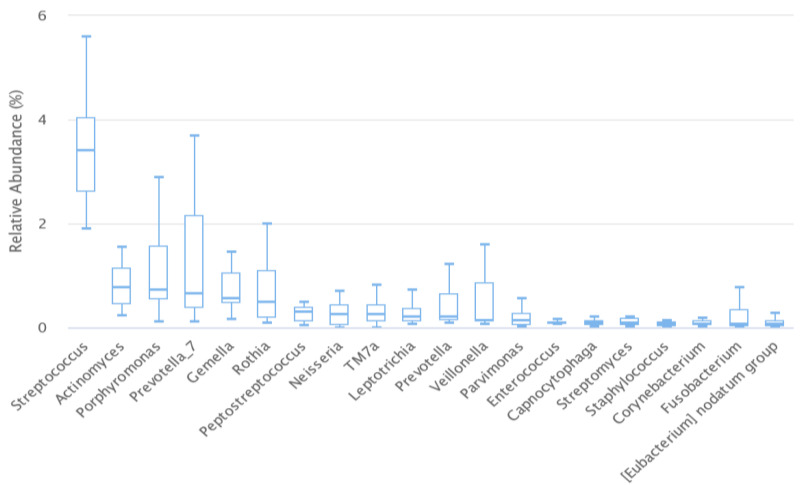
Distribution of dominant genera in saliva samples of all participants.

**Figure 4 biomedicines-11-01012-f004:**
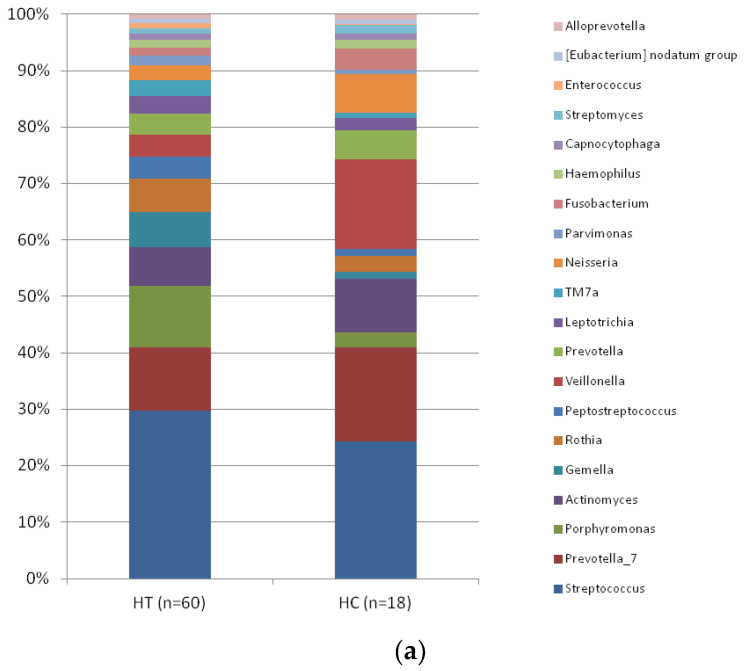
(**a**) The distributions of the predominant bacteria at the genus level between HT and HC groups. (HT: Hashimoto’s thyroiditis. HC: Healthy control). (**b**) The distributions of the predominant bacteria at the genus level between LT-HT, LT + HT, and HC groups. (HT: Hashimoto’s thyroiditis. LT + HT: Euthyroid HT treated with levothyroxine (L-T4). LT-HT: euthyroid HT without L-T4 therapy. HC: Healthy control).

**Table 1 biomedicines-11-01012-t001:** Demographic characteristics for all participants (Mean + SD *).

	LT – HT (*n* = 32)	LT + HT (*n* = 28)	HC (*n* = 18)	*p* **
Age (years)	39.4 ± 4.52	39.8 ± 6.04	39.1 ± 5.8	0.893
BMI (kg/m^2^)	21.6 ± 2.1	22.4 ± 1.9	20.8 ± 2.3	0.914
TSH (mU/L)	1.86 ± 1.81	2.12 ± 1.94	1.61 ± 1.35	0.785
TPOAb (IU/mL)	265 ± 224	342 ± 274	7.8 ± 4.9	<0.0001

* Values are presented as mean ± standard deviation. ** The Kruskal Wallis test. TPOAb: thyroid peroxidase antibody. TSH: thyroid-stimulating hormone. HT: Hashimoto’s thyroiditis. LT + HT: Euthyroid HT treated with levothyroxine (L-T4). LT – HT: euthyroid HT without L-T4 therapy. HC: Healthy control.

**Table 2 biomedicines-11-01012-t002:** Alpha-diversity analysis of bacterial communities between groups (Mean + SD*).

Saliva Samples	LT – HT (n = 32)	LT + HT (n = 28)	HC (n = 18)	*p* **
Simpson	0.89 ± 0.04	0.91 ± 0.03	0.87 ± 0.05	0.952
Shannon	3.71 ± 2.12	3.68 ± 1.94	3.73 ± 2.03	0.874

* Values are presented as mean ± standard deviation. ** The Kruskal Wallis test. HT: Hashimoto’s thyroiditis. LT + HT: Euthyroid HT treated with levothyroxine (L-T4). LT – HT: euthyroid HT without L-T4 therapy. HC: Healthy control.

**Table 3 biomedicines-11-01012-t003:** Distribution of different microbiota reports on HT.

References	Sample Type	Increased in HT	Decreased in HT
Wang et al. [19]	Saliva and Stool	*Gammaproteobacteria* and *Prevotella*	*Firmicutes*, *Leptotrichiace*, and *Actinobacteria*
Zhao et al. [30]	Stool	*Blautia*, *Roseburia*, *Ruminococcus_torques_group*, *Romboutsia*, *Dorea, Fusicatenibacter*, and *Eubacterium_hallii_group*	*Faecalibacterium*, *Bacteroides*, *Prevotella_9*, and *Lachnoclostridium*
Ishaq et al. [31]	Stool	*Firmicutes*, *Bacteroidetes*, *Actinobacteria*, *Verrucomicrobia*, *Fusobacteria Escherichia-Shigella*, and *Parasutterella*	*Proteobacteria*, *Prevotella*_9 and *Dialister*
Liu et al. [32]	Stool	*Bacteroidetes*, *Actinobacteria*, *Fusobacteria*, *Proteobacteria*, and *Alistipes*	*Firmicutes*, *Verrucomicrobia*, and *Faecalibacterium*
Zhao et al. [34]	Stool	*Proteobacteria*, *Actinobacteria*, *Erysipelotrichia*, *Cyanobacteria*, and *Ruminococcus*_2	*Peptostreptococcaceae*, *Bacillaceae*, and *Matophyaceae*
In this study	Saliva	*Firmicutes*, *Bacteroidota*, *Actinobacteriota*, *Patescibacteria*, *Fusobacteriota*, *Spirochaetota*, *Enterococcus*, and *Staphylococcus*	*Proteobacteria*, *Campylobacterota*, and *Bacillus*

## Data Availability

No new data were created.

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
