# Peer review of "Oral Microbiota Signatures in the Pathogenesis of Euthyroid Hashimoto’s Thyroiditis"

_biomedicines, 2023, doi:10.3390/biomedicines11041012_

Round 1

Reviewer 1 Report

It is acceptable in current form.

Author Response

Dear Reviewer,

Thank you very much for your valuable comments and contribution. All comments have been carefully checked and edited on the manuscript.

You can also see our response below.

Thank you, Regards

Assoc. Prof. Mehmet Demirci

Response to Reviewer 1:

It is acceptable in current form.

Response: Thank you for your valuable comments and contributions. 

Reviewer 2 Report

The article is very interesting from a clinical and scientific point of view. Especially for dentists, because changes in the oral microbiome can not only be the cause of exacerbation of autoimmune diseases, but also disturb the homeostasis of the oral cavity and be the cause of diseases.

Author Response

Dear Reviewer,

Thank you very much for your valuable comments and contribution. All comments have been carefully checked and edited on the manuscript.

You can also see our response below.

Thank you, Regards

Assoc. Prof. Mehmet Demirci

Response to Reviewer 2:

The article is very interesting from a clinical and scientific point of view. Especially for dentists, because changes in the oral microbiome can not only be the cause of exacerbation of autoimmune diseases, but also disturb the homeostasis of the oral cavity and be the cause of diseases.

Response: Thank you for your valuable comments and contributions. 

Reviewer 3 Report

The original article entitled ``Oral microbiota signatures in the pathogenesis of euthyroid Hashimoto's thyroiditis`` is well written. The results are well presented.

 Minor revision

Discussions could be improved. Please insert in discussion a table with literature data synthesis about this subject. This is very important because this table will show the contribution of this study.

There are minor language errors. 

The figures and tables legend must include the explanation for all acronyms.

Author Response

Dear Reviewer,

Thank you very much for your valuable comments and contribution. All comments have been carefully checked and edited on the manuscript.

You can also see our response below.

Thank you, Regards

Assoc. Prof. Mehmet Demirci

Response to Reviewer 3:

The original article entitled ``Oral microbiota signatures in the pathogenesis of euthyroid Hashimoto's thyroiditis`` is well written. The results are well presented.

Response: Thank you for your valuable comments and contributions. 

Minor revision

Discussions could be improved. Please insert in discussion a table with literature data synthesis about this subject. This is very important because this table will show the contribution of this study.

Response: Thank you for your valuable comments and contributions.  On your suggestion, The correction was carried out and Table 3 was added in the manuscript.

There are minor language errors.

Response: Thank you for your valuable comments and contributions. On your suggestion, The corrections were made to the manuscript. Editing service was taken from native english speaker.

The figures and tables legend must include the explanation for all acronyms.

Response: Thank you for your valuable comments and contributions. On your suggestion, all acronyms were inserted below figures and tables in the manuscript.

Reviewer 4 Report

The paper by M.G. Erdem et al. is focused on the examination of oral microbiota patients with Hashimoto's thyroiditis. The study, although it employs advanced sequencing techniques, has many limitations including relatively small number of studied subjects and lacking tests for numerous factors other than Hashimoto’s disease that could possibly affect the status of patient’s immune system (e.g. CRP level). More importantly, no data on the history of the disease were collected: i.e. when the disease was diagnosed, what was thyroid function then, when the treatment was started and so. Nonetheless, I agree with the authors that these limitations are not crucial as the study is preliminary in its nature. However, a better description of these limitations would be appreciated.
My main remark is that the authors could not decide whether they believe that the altered oral microbiota possibly stimulate the development of the disease (as they suggest by some data they refer to in the introduction or the discussion) or rather it is vice versa – Hashimoto’s disease leads to some changes in the oral biota – which the authors seems to conclude. This uncertainty could be addressed by a more thoughtful study design, especially more attention paid to the course of the disease in studied patients. For now, it would be advisable to describe these possibilities in the paper and to not mix them in lines of reasoning presented in the introduction and the discussion. And, accordingly, conclusions should be drawn with care.

 I am not familiar with sequencing techniques used in the study but their description seems to adequate.

Specific remarks:

The abstract ends with the following conclusion: “Therefore, revealing the core oral microbiota and long-term follow-up of the HT process by conducting extensive and multicenter studies will provide essential data for understanding the pathogenesis of the disease.” – It seems far too strong (see my general remrks). We can’t be sure about that, so maybe ‘might provide some important data’ would be more appropriate.

Lines 45-46: “patients often report changes in … thyroid functions as a result of dietary changes”. Really? An average patient has little knowledge of thyroid function and consequences of its disturbances. Of course, iodine supply influences thyroid function. As well as natural goitrogens. But even physicians are usually unable to enumerate foods that contain goitrogens.  

The aim of the study as defined at the end the introduction suggests there was one uniform group of female patients with Hashimoto’s disease treated with levothyroxine plus a control group. Actually, there were two groups of patients with chronic thyroiditis: treated and non-treated. Please rephrase the aim of the study accordingly.

Line 71: “The following criteria were checked in the entire study groups and these participants were excluded from the study”. Well, I guess the meaning but it is an example of terrible language structure – please rephrase it.

Line 78: “HT diagnosis was based on the thyroid antibodies and thyroid ultrasonography results”. Please be more specific: what antibodies and what features of ultrasound image were considered. Was a positive ultrasound image enough for the diagnosis (or positive antibodies lab test) or rather both were needed?

Author Response

Dear Reviewer,

Thank you very much for your valuable comments and contribution. All comments have been carefully checked and edited on the manuscript.

You can also see our response below.

Thank you, Regards

Assoc. Prof. Mehmet Demirci

Response to Reviewer 4:

The paper by M.G. Erdem et al. is focused on the examination of oral microbiota patients with Hashimoto's thyroiditis. The study, although it employs advanced sequencing techniques, has many limitations including relatively small number of studied subjects and lacking tests for numerous factors other than Hashimoto’s disease that could possibly affect the status of patient’s immune system (e.g. CRP level). More importantly, no data on the history of the disease were collected: i.e. when the disease was diagnosed, what was thyroid function then, when the treatment was started and so. Nonetheless, I agree with the authors that these limitations are not crucial as the study is preliminary in its nature. However, a better description of these limitations would be appreciated.
My main remark is that the authors could not decide whether they believe that the altered oral microbiota possibly stimulate the development of the disease (as they suggest by some data they refer to in the introduction or the discussion) or rather it is vice versa – Hashimoto’s disease leads to some changes in the oral biota – which the authors seems to conclude. This uncertainty could be addressed by a more thoughtful study design, especially more attention paid to the course of the disease in studied patients. For now, it would be advisable to describe these possibilities in the paper and to not mix them in lines of reasoning presented in the introduction and the discussion. And, accordingly, conclusions should be drawn with care.

Response: Thank you for your valuable comments and contributions. On your suggestion, The correction was carried out in the manuscript. Limitation part and conclusion section were revised.

 I am not familiar with sequencing techniques used in the study but their description seems to adequate.

Response: Thank you for your valuable comments and contributions. 

Specific remarks:

The abstract ends with the following conclusion: “Therefore, revealing the core oral microbiota and long-term follow-up of the HT process by conducting extensive and multicenter studies will provide essential data for understanding the pathogenesis of the disease.” – It seems far too strong (see my general remrks). We can’t be sure about that, so maybe ‘might provide some important data’ would be more appropriate.

Response: Thank you for your valuable comments and contributions. On your suggestion, The correction was carried out in the manuscript at abstract and conclusion parts

Lines 45-46: “patients often report changes in … thyroid functions as a result of dietary changes”. Really? An average patient has little knowledge of thyroid function and consequences of its disturbances. Of course, iodine supply influences thyroid function. As well as natural goitrogens. But even physicians are usually unable to enumerate foods that contain goitrogens.  

Response: Thank you for your valuable comments and contributions. Although you have reported that this is not case in real life, we could not revise it because we had to rely on the references.

The aim of the study as defined at the end the introduction suggests there was one uniform group of female patients with Hashimoto’s disease treated with levothyroxine plus a control group. Actually, there were two groups of patients with chronic thyroiditis: treated and non-treated. Please rephrase the aim of the study accordingly.

Response: Thank you for your valuable comments and contributions. On your suggestion, The correction was carried out in the manuscript. Groups explained more clearly.

Line 71: “The following criteria were checked in the entire study groups and these participants were excluded from the study”. Well, I guess the meaning but it is an example of terrible language structure – please rephrase it.

Response: Thank you for your valuable comments and contributions. On your suggestion, The correction was carried out in the manuscript

Line 78: “HT diagnosis was based on the thyroid antibodies and thyroid ultrasonography results”. Please be more specific: what antibodies and what features of ultrasound image were considered. Was a positive ultrasound image enough for the diagnosis (or positive antibodies lab test) or rather both were needed?

Response: Thank you for your valuable comments and contributions. On your suggestion, The correction was carried out in the manuscript
